# Wide range of metabolic adaptations to the acquisition of the Calvin cycle revealed by comparison of microbial genomes

**Johannes Asplund-Samuelsson, Elton P. Hudson** [ID] *

Science for Life Laboratory, School of Engineering Sciences in Chemistry, Biotechnology and Health, KTH Royal Institute of Technology, Solna, Sweden

* huds@kth.se

**Data Availability Statement:** Translated open reading frames and their Pfam family and Enzyme Commission number annotations are available

## Abstract

Knowledge of the genetic basis for autotrophic metabolism is valuable since it relates to both the emergence of life and to the metabolic engineering challenge of incorporating $CO_2$ as a potential substrate for biorefining. The most common $CO_2$ fixation pathway is the Calvin cycle, which utilizes Rubisco and phosphoribulokinase enzymes. We searched thousands of microbial genomes and found that 6.0% contained the Calvin cycle. We then contrasted the genomes of Calvin cycle-positive, non-cyanobacterial microbes and their closest relatives by enrichment analysis, ancestral character estimation, and random forest machine learning, to explore genetic adaptations associated with acquisition of the Calvin cycle. The Calvin cycle overlaps with the pentose phosphate pathway and glycolysis, and we could confirm positive associations with fructose-1,6-bisphosphatase, aldolase, and transketolase, constituting a conserved operon, as well as ribulose-phosphate 3-epimerase, ribose-5-phosphate isomerase, and phosphoglycerate kinase. Additionally, carbohydrate storage enzymes, carboxysome proteins (that raise $CO_2$ concentration around Rubisco), and Rubisco activases CbbQ and CbbX accompanied the Calvin cycle. Photorespiration did not appear to be adapted specifically for the Calvin cycle in the non-cyanobacterial microbes under study. Our results suggest that chemoautotrophy in Calvin cycle-positive organisms was commonly enabled by hydrogenase, and less commonly ammonia monooxygenase (nitrification). The enrichment of specific DNA-binding domains indicated Calvin-cycle associated genetic regulation. Metabolic regulatory adaptations were illustrated by negative correlation to AraC and the enzyme arabinose-5-phosphate isomerase, which suggests a downregulation of the metabolite arabinose-5-phosphate, which may interfere with the Calvin cycle through enzyme inhibition and substrate competition. Certain domains of unknown function that were found to be important in the analysis may indicate yet unknown regulatory mechanisms in Calvin cycle-utilizing microbes. Our gene ranking provides targets for experiments seeking to improve $CO_2$ fixation, or engineer novel $CO_2$-fixing organisms.

from Figshare (DOI 10.6084/m9.figshare.13013309). All other relevant data are within the manuscript and its Supporting information files.

**Funding:** This project was supported by the Swedish Research Council Vetenskapsrådet (https://www.vr.se/) through grant number 2016-06160 to EPH, the Swedish Strategic Research Foundation SSF (https://strategiska.se/) through grant number ARC19-0051 to EPH, and by Novo Nordisk Fonden (https://novonordiskfonden.dk/) through grant number NNF20OC0061469 to EPH. The funders had no role in study design, data collection and analysis, decision to publish, or preparation of the manuscript.

**Competing interests:** The authors have declared that no competing interests exist.

## Author summary

Rising carbon dioxide levels driving climate change prompts us to embrace sustainable resources, such as autotrophic microbes that produce biomass or chemicals by consuming carbon dioxide. As genetic engineering of natural autotrophs is challenging, it is of interest to engineer autotrophy in more pliable microbial species, such as *Escherichia coli*. We contrasted 1,020 genomes of microbes carrying the most widespread carbon dioxide fixation pathway, the Calvin cycle, to genomes of closest relatives lacking this pathway. This comparison identified and ranked genetic adaptations that may enable Calvin cycle operation. This list of adaptations sheds light on the evolution of autotrophy and represents a recipe for an autotrophic microbe, which can aid genetic engineers in improving autotrophs or creating them from scratch.

## Introduction

Organisms that produce biomass by fixation of $CO_2$ are classified as autotrophic. As atmospheric $CO_2$ levels rise, autotrophs offer attractive ecological and biotechnological routes to climate change mitigation and sustainable biomanufacturing. Autotrophs such as Cyanobacteria, algae, and plants already serve as primary producers in most ecosystems. Emphasizing the central role of autotrophic metabolism in evolution and life, the last universal common ancestor possessed the Wood-Ljungdahl pathway for $CO_2$ fixation [1], possibly in combination with the reductive tri-carboxylic acid (TCA) cycle and the reductive glycine pathway [2]. These three ancient $CO_2$ fixation pathways were later accompanied by the dicarboxylate/4-hydroxybutyrate cycle [3], the 3-hydroxypropionate/4-hydroxybutyrate cycle [4], the 3-hydroxypropionate bicycle [5,6], and the Calvin-Benson-Bassham (CBB) cycle [7].

The CBB cycle, or Calvin cycle, is the most common $CO_2$ fixation pathway in living organisms [8,9]. The Calvin cycle is distinguished by phosphoribulokinase (Prk), which phosphorylates the phosphosugar ribulose-5-phosphate to ribulose-1,5-bisphosphate using ATP, and ribulose bisphosphate carboxylase/oxygenase (Rubisco), which carboxylates ribulose-1,5-bisphosphate with $CO_2$, thereby generating two molecules of 3-phosphoglycerate. Rubisco oxygenation of ribulose-1,5-bisphosphate generates toxic 2-phosphoglycolate, prompting recycling through the photorespiration pathway, reducing carbon yield [10]. While 3-phosphoglycerate connects to glycolysis/gluconeogenesis and the TCA cycle, ribulose-5-phosphate connects to the pentose phosphate pathway (PPP) and ribonucleotide synthesis [11]. Accordingly, the Calvin cycle is also named the reductive pentose phosphate cycle. Rubisco evolved from a methionine salvage enzyme [12] over 2.9 billion years ago [13], before the great oxygenation event [14]. Research has explored Rubisco evolution [15–19] and biochemistry [20–23], identifying several forms [24–26]. Form I (Cyanobacteria, Proteobacteria, other Bacteria, algae, and plants), form II (Proteobacteria, Archaea, and dinoflagellates), and form III (Archaea) Rubiscos fix $CO_2$, while form IV "Rubisco-like" proteins (RLPs) retain the earlier methionine salvage role.

The overlap of the Calvin cycle with the PPP has inspired attempts to grant Rubisco-catalyzed $CO_2$ fixation to heterotrophs with minimal insertion of heterologous genes. For example, transformation with just Prk and Rubisco reduces carbon loss during fermentation in yeast [27] and *E. coli* [28]. In a series of reports, it was recently shown that introduction of Prk and Rubisco, followed by selected severing from glycolysis metabolism and directed evolution, could result in complete autotrophic generation of biomass from $CO_2$ in *E. coli* [29,30].

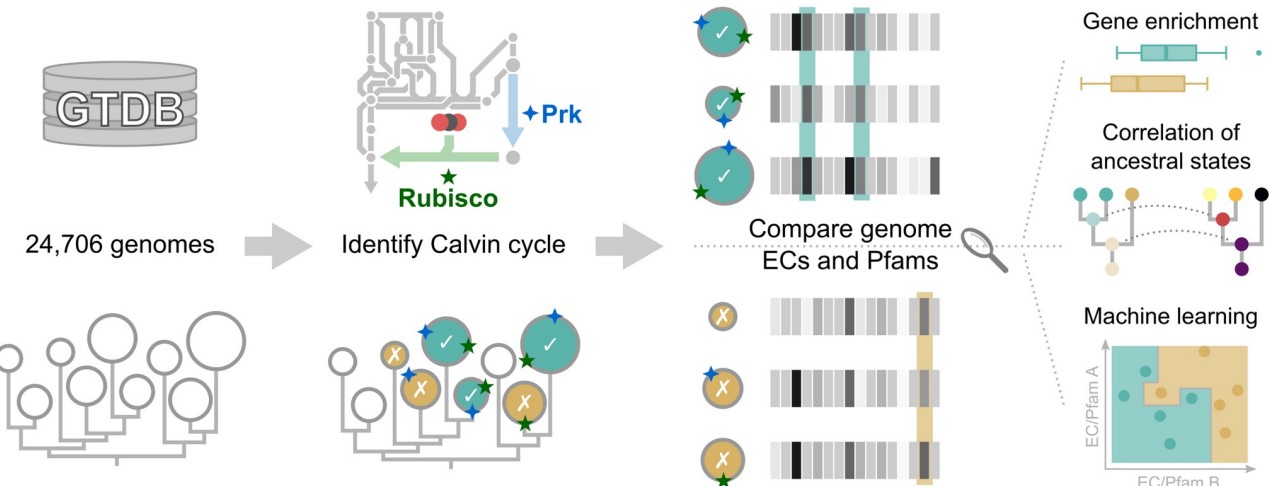

**Fig 1. Sequence-based analysis identifies genetic adaptations unique to Calvin cycle-containing genomes.** Bacterial and archaeal genomes from the Genome Taxonomy Database (GTDB) were subjected to a Hidden Markov Model-based homology search for Prk and Rubisco, which identified Calvin cycle-positive genomes. The Calvin cycle-positive genomes were contrasted as a collective against their closest relatives to identify genes, *i.e.* Enzyme Commission (EC) numbers and Pfams, associated with the Calvin cycle via three statistical comparison methods. First, enrichment identified genes that were generally depleted or enriched in Calvin cycle-positive genomes using a Wilcoxon rank sum test. Second, phylogenetics-based ancestral character estimation was used on subtrees in order to correlate the emergence of the Calvin cycle to other genes. Third, a random forest machine learning algorithm was employed to distinguish between Calvin cycle-positive and Calvin cycle-negative genomes based on other genes that were thereby ranked according to their importance in the classification task.

Growth of *E. coli* required 150–250 generations of adaptive evolution to enable stable Calvin cycle operation. Carbon retention within the cycle was enabled by mutations suppressing glucose-phosphate isomerase, which initiates glycogen synthesis, and ribose-phosphate-diphosphokinase, which diverts pentoses to nucleotide metabolism [30,31], consistent with branch point reactions being crucial for autocatalytic cycle stability [32]. In light of these reports, a comparison of microbial genomes with and without the Calvin cycle could also expose metabolic adaptations responding to the acquisition of Prk and Rubisco, which may aid engineering new autotrophs.

Horizontal gene transfer (HGT) processes raise the question of how genomes accommodate the Calvin cycle in nature. For example, HGT of Rubisco has occurred within Proteobacteria, and from Proteobacteria to Cyanobacteria and plastids [33]. Rubisco was also horizontally acquired together with Prk in candidate phyla radiation Bacteria and DPANN Archaea [34], and together with the regulator CbbR and the activase chaperone CbbQ in *Rhodobacter capsulatus* [35]. Rubisco HGT has also been reported in acid mine drainage microbiomes [36]. Interestingly, Calvin cycle operons locate to plasmids in *Ralstonia eutropha* [37] and *Oligotropha carboxidovorans* [38]. *R. eutropha* has a second, nearly identical operon encoding the Calvin cycle enzymes situated on the chromosome [39], which may reflect a horizontal transfer via plasmid caught in progress.

Here, we sought to shed light on the natural adaptations surrounding autotrophic metabolism. We contrasted 1,020 archaeal and bacterial genomes possessing the Calvin cycle with genomes from the 1,020 closest relatives without the Calvin cycle and thereby identified a range of Calvin cycle-associated adaptations (Fig 1). The adaptations reported here may also inspire future metabolic engineering initiatives aiming to generate artificial Calvin cycle-utilizing organisms.

## Results and discussion

### Genomes were classified as Calvin cycle-positive or -negative

We first sought a way to classify genomes as CBB-positive and CBB-negative. Our approach was to search for the unique Calvin cycle genes Rubisco (large subunit) and Prk in the 24,706 bacterial and archaeal genomes of the Genome Taxonomy Database (GTDB; https://gtdb. ecogenomic.org/). Using Hidden Markov Model (HMM) profiles, we detected 2,348 Rubisco large subunit and 4,828 Prk sequences, in 2,141 and 4,284 genomes, respectively. We did not include the Rubisco small subunit in the search since the small subunit lacks catalytic activity and is only present in form I Rubisco [40]. Phylogenetic analysis showed that all Rubisco forms (I-III) were detected and also that Rubisco-like proteins (form IV), which do not catalyze $CO_2$ fixation, were excluded (S1 Fig). We assumed that the presence of both Rubisco and Prk in 1,490 of the genomes (6.0%) indicated a complete Calvin cycle and thus CBB-positive classification. The fraction of CBB-positive genomes was lower than the 7.2% in KEGG (https://www.kegg.jp/), which could be because incomplete sequencing and metagenome assemblies caused some genomes to be incorrectly identified as CBB-negative. For example, Cyanobacteria are CBB-positive, but 184 (28%) of 654 cyanobacterial genomes were classified as CBB-negative. Cyanobacterial CBB-positive and CBB-negative genomes showed 98.8% and 82.8% median completeness ($p \approx 5.8 \cdot 10^{-33}$, Wilcoxon rank sum test), and 94% of the $\geq 99\%$ complete genomes were CBB-positive, while just 60% of the $< 99\%$ complete genomes were CBB-positive. If underestimated by 28%, as in Cyanobacteria, the true fraction of CBB-positive genomes was 8.4%. Additionally, the carboxydotrophic genus *Hydrogenophaga* [41] fixes carbon using the Calvin cycle [42,43], but *H. pseudoflava* was 99.6% complete and CBB-negative, due to missing Prk, while *H. flava* was 98.8% complete and CBB-positive. Here the open reading frame (ORF) identification was flawed, since the Prk HMM yielded a hit (accession WP_066156609.1) among the *H. pseudoflava* RefSeq ORFs (accession GCF_001592285.1). Furthermore, incorrectly CBB-positive genomes may appear due to HMM hits scoring close to the threshold (see Materials and methods). Among 50 randomly selected CBB-positive genomes (S1 Table), 43 were mentioned in the literature, with 30 genomes likely to be true CBB-positive examples, and only one a likely false CBB-positive human pathogen (*Mycolicibacterium mageritense*). The limitations imposed by incorrect classifications of individual genomes were relieved by using global comparison methods.

Cyanobacteria are very different from other Bacteria due to an ancient evolutionary emergence [44,45], and their inclusion could potentially bias a comparison between CBB-positive and closely related, CBB-negative genomes. For example, median distances from CBB-positive genomes to their closest CBB-negative relatives were 0.75 for Cyanobacteria (n = 470) and 0.14 for non-cyanobacterial microbes (n = 1,020; $p \approx 1.5 \cdot 10^{-83}$, Wilcoxon rank sum test), indicating significant divergence of Cyanobacteria from their most closely related, non-CBB genomes. The closest CBB-negative relatives of Cyanobacteria include false CBB-negative Cyanobacteria (154 genomes), but mostly Firmicutes (313 genomes), primarily of class Bacilli (176 genomes). While Cyanobacteria are a highly relevant biotechnological platform, including them here would bias the dataset, so that any discriminating genetic differences would be specific to separating the large number of Cyanobacteria from all other bacteria instead of focusing on early adaptations specific to acquisition of the Calvin cycle. Additionally, the distant "close relatives" of Cyanobacteria, *i.e.* Firmicutes, could introduce additional noise from their specific genes compared to the rest of the dataset. To maintain the focus on early adaptations, Cyanobacteria were excluded from further analysis, leaving 1,020 CBB-positive genomes. We note that genetic differences among Cyanobacteria have been investigated previously [46].

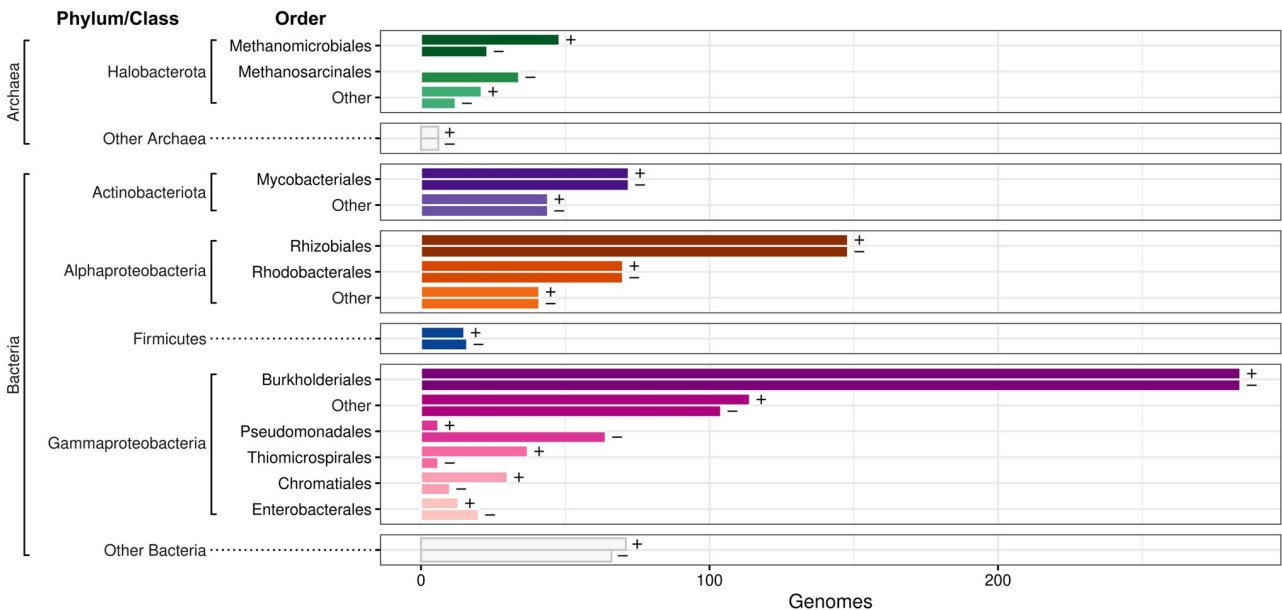

**Fig 2. The Calvin cycle is present in a diverse range of Bacteria and Archaea.** Bars display the taxonomic distribution of 1,020 CBB-positive (+) and 1,020 CBB-negative (-) genomes analyzed in this study. Bars are grouped by phylum, or class for Proteobacteria. The orders with most members have separate bars, while other organisms are aggregated under the "Other" labels.

The 945 CBB-positive bacterial genomes were mainly from the orders Burkholderiales and Rhizobiales, other Alpha- and Gammaproteobacteria, and some Actinobacteriota and Firmicutes. The 75 CBB-positive archaeal genomes were mainly from the Halobacterota orders Methanomicrobiales, Archaeoglobales, and Methanotrichales, and may harbor the CBB-like reductive hexulose-phosphate pathway studied in methanogenic Archaea [47]. We picked an equal number (1,020) of CBB-negative genomes with the shortest possible phylogenetic distance to the CBB-positive genomes to serve as a contrasting dataset with similar taxonomic distribution (Fig 2).

### Three complementary statistical analyses to rank genes for their relevance to the acquisition of the Calvin cycle

From the 2,040 selected genomes (ORFs available at https://doi.org/10.6084/m9.figshare. 13013309) we identified genetic adaptations that specifically accompany the Calvin cycle using three methods: enrichment analysis, ancestral character estimation (ACE), and random forest machine learning. We used copy numbers of Pfam domains (https://pfam.xfam.org/) and copy numbers of enzymes in these genomes as "genetic features," to be investigated (12,703 features; ORF annotations available at https://doi.org/10.6084/m9.figshare.13013309, and copy numbers provided in S1 Dataset). Enzymes were assigned Enzyme Commission (EC) numbers by DeepEC [48]. We will refer to these genetic features simply as "genes" in the remainder of the text. The enrichment analysis compared gene copy number distributions between CBB-positive and CBB-negative genomes using a Wilcoxon rank sum test (S2 Dataset). ACE "rewinded" evolution by estimating the ancestral CBB status (positive or negative) and copy number of genes in phylogenetic tree clades that we call subtrees. The estimations of ancestral CBB status and gene copy number were then correlated (S3 Dataset). Finally, the random forest analysis encompassed training a machine learning classifier to distinguish between CBB-positive and CBB-negative genomes. The output is an importance value for 1,200 genes,

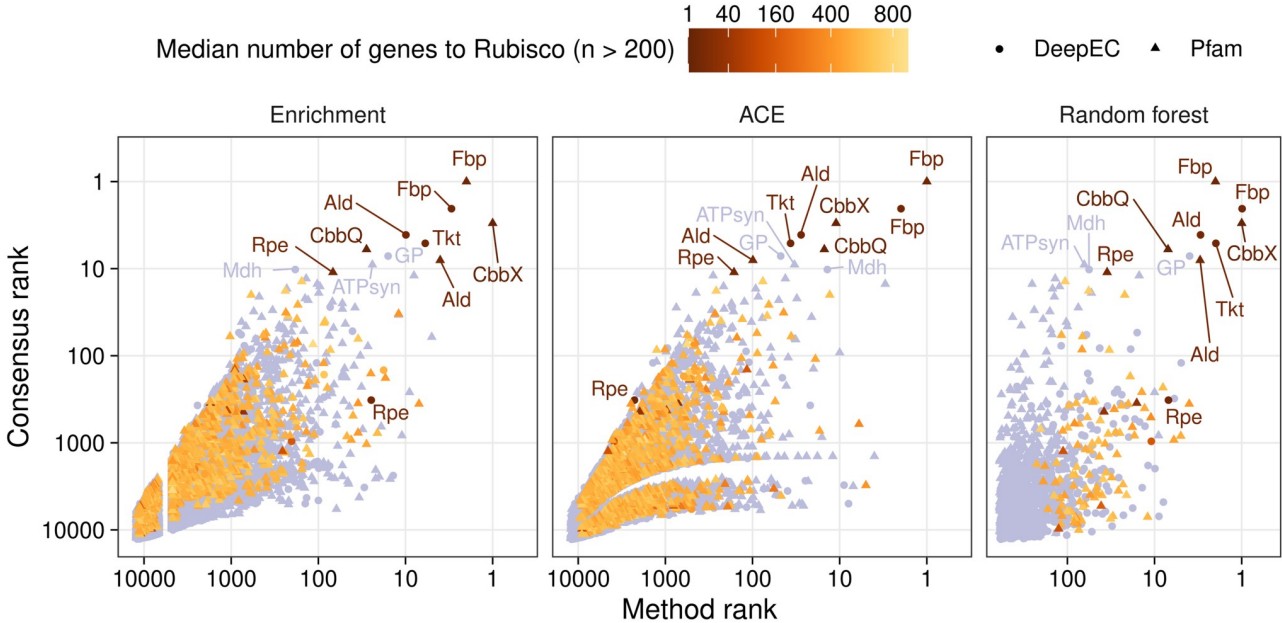

**Fig 3. Three methods for identifying and ranking the importance of genes distinguishing Calvin cycle-positive genomes from relatives cooperate to yield a consensus ranking.** The rank of genes, *i.e.* Enzyme Commission numbers (EC) or Pfam families, within each method (x-axes, logarithmic scale) is plotted against the consensus rank from all three methods (y-axis, logarithmic scale). The orange color intensity (square root scale) indicates the median distance between the gene and Rubisco in number of genes in CBB-positive genomes (S6 Dataset), if the gene was found on the same DNA strand as Rubisco more than 200 times. Genes detected 200 times or fewer on the same DNA strand as Rubisco are shown in light purple. Note that ECs and Pfams were ranked separately in the random forest analysis and thereby each rank is shared by one EC and one Pfam. The random forest analysis included only 1,200 genes due to so-called feature selection preceding ranking (see Materials and methods). The gap between ranks 4,714 and 6,824 in the enrichment analysis is due to 2,110 genes sharing the same $q$ value used for ranking (S2 Dataset). Abbreviations: Ald, fructose-bisphosphate aldolase (EC 4.1.2.13, PF01116); ATPsyn, ATP synthase (PF02823); CbbQ, Rubisco activase CbbQ (PF08406); CbbX, Rubisco activase CbbX (PF17866, "AAA_lid_6"); Fbp, fructose-1,6-bisphosphatase (EC 3.1.3.11, PF00316); GP, glycogen phosphorylase (EC 2.4.1.1); Mdh, malate dehydrogenase (EC 1.1.5.4); Rpe, ribulose-phosphate 3-epimerase (EC 5.1.3.1, PF00834); Tkt, transketolase (EC 2.2.1.1).

selected through logistic regression before training, that relates to their contribution to this classifier (S4 Dataset). After analysis with the three methods, genes were ranked by seeking low enrichment $q$ values, high weighted sum of absolute ACE correlations, and high random forest importance (Fig 3). The rankings correlated best between the ACE and random forest analyses (Spearman $r \approx 0.44$, $p \approx 1.0 \cdot 10^{-57}$, n = 1,192), followed by enrichment and random forest ($r \approx 0.23$, $p \approx 4.5 \cdot 10^{-16}$, n = 1,194), and ACE and enrichment ($r \approx 0.20$, $p \approx 1.8 \cdot 10^{-126}$, n = 11,731). The agreement in rankings between the ACE and random forest analyses may reflect their ability to emphasize positive and negative associations at the same time.

Using three different methods allowed us to avoid outliers in any one particular method. The three methods also ensured that every gene and aspect of adaptation was probed thoroughly. For example, while machine learning should be the most powerful way to rank genes, it is recommended to filter the input through feature selection to improve performance, reduce training time, and avoid overfitting. In our case the random forest probed only the 1,200 most promising genes, just a fraction of the total feature set. Furthermore, the random forest machine learning model provides a ranking through the so-called feature importance values, but it does not provide straightforward information on how it reaches those conclusions. Therefore, we needed the enrichment analysis to determine if the copy numbers of specific genes were generally higher or lower in CBB-positive genomes. We also needed the ACE analysis to determine if genes showed positive or negative correlation to the emergence of the Calvin cycle within specific microbial groups. The ranks calculated by the three methods were

added together and then ranked again, which gave a final *consensus rank* that included 12,501 genes (Fig 3, Table 1, S5 Dataset). A top consensus rank indicated that the three methods were in agreement, thereby identifying the most prominent genetic changes associated with the Calvin cycle. From here on, we only report the consensus rank in the text.

Before examining the biological results, we first briefly discuss the limitations of the three individual statistical comparison methods. The enrichment analysis identified 878 genes significantly enriched or depleted in CBB-positive genomes ($q < 0.05$), with $\log_2$ ratios of CBB-positive to CBB-negative copy numbers ranging from -6.0 to 6.3 (95% of values -2.6 to 2.2). Genes with low $q$ values contributed to a prominent consensus rank. Rubisco activase CbbX (PF17866) had the lowest $q$ value, *i.e.* $6.8 \cdot 10^{-22}$, indicating clear separation between CBB-positive (0.77 copies) and CBB-negative genomes (0.40 copies). At the other end of the significance spectrum, the Major Facilitator Superfamily (PF07690) showed $q$ value 0.038 and the smallest significant absolute $\log_2$ ratio (0.012) between CBB-positive (38.6 copies) and CBB-negative genomes (39.0 copies).

The ACE analysis was influenced by the structure of the phylogenetic trees, so that Calvin cycle status signal was lost as leaf node patterns increased in complexity. For example, ancestral nodes had equal likelihood of being CBB-positive and CBB-negative in subtree 1 when combined with other subtrees (Fig 4A). However, when subtree 1 was analyzed separately the ancestral characters became visible (Fig 4C). Therefore, we correlated the Calvin cycle and other genes using the twelve subtrees. Ultimately, subtree 1 (Fig 4C) yielded a significant correlation for 25% of the 7,265 genes in those organisms. At the extremes, Archaea yielded 2,217 significant genes out of 5,672 (39%), while subtree 3 yielded none out of 6,600. The second highest number of significant genes among bacteria was 13% in subtree 5, indicating that Archaea and bacterial subtree 1 dominated the contribution from the ACE analysis in the consensus ranking.

The random forest analysis yielded a machine learning classifier that labeled genomes as CBB-positive or CBB-negative with 72.9% accuracy using ECs and 76.5% accuracy using Pfams, compared to the expected accuracy of 50% if picking labels at random. For comparison, it has been shown that a random forest can achieve 88% accuracy in predicting photosynthetic proteins based on gene neighborhood [49], a tree-based classifier can achieve 86% accuracy (94% with a k-nearest neighbor model) in predicting the recombination status of HIV genomes [50], and a support vector machine can achieve 87% accuracy in classifying bacteria as pathogenic or not based on their proteomes [51]. The accuracy achieved in the present study was likely limited by false negative genomes, genomes in the process of adapting to recent loss or gain of the Calvin cycle, a low amount of training data, and limited ability of the random forest to focus on relevant aspects of the data. Nevertheless, our algorithm appeared to identify relevant biological information distinguishing CBB-positive genomes from those without the Calvin cycle (discussed below).

Below we report on what we identified as the most prominent, interesting, and relevant biological patterns in the dataset. We consider adaptations in the top 10%, *i.e.* the top 1250 genes, to have good consensus ranks, while others are poor, but the focus is mainly on the top 200 genes. There are other adaptations with a narrower scope that are not discussed here, but can be found in the Supporting information.

## Core Calvin cycle enzyme genes are generally enriched in genomes with Rubisco and Prk

Fructose-1,6-bisphosphatase (EC 3.1.3.11), aldolase (EC 4.1.2.13), and transketolase (EC 2.2.1.1) each ranked within the top five genes (Table 1), thus highlighting the Calvin cycle's

**Table 1. Consensus rank of genetic adaptations to the Calvin cycle.**

| Rank | Gene | CBB⁺ | CBB⁻ | ACE r | Imp. | Description |
|---|---|---|---|---|---|---|
| 2 | EC 3.1.3.11 | 1.05 | 0.69 | 0.29 | 0.0200 | fructose-bisphosphatase |
| 4 | EC 4.1.2.13 | 0.99 | 0.69 | 0.2 | 0.0149 | fructose-bisphosphate aldolase |
| 5 | EC 2.2.1.1 | 1.33 | 0.98 | 0.19 | 0.0164 | transketolase |
| 7 | EC 2.4.1.1 | 0.69 | 0.46 | 0.19 | 0.0104 | glycogen phosphorylase |
| 10 | EC 1.1.5.4 | 0.11 | 0.19 | 0.3 | 0.0042 | malate dehydrogenase (quinone) |
| 27 | EC 1.4.4.2 | 1.04 | 0.87 | 0.16 | NA | glycine dehydrogenase (aminomethyl-transferring) |
| 51 | EC 2.7.7.27 | 0.71 | 0.52 | 0.14 | NA | glucose-1-phosphate adenylyltransferase |
| 53 | EC 3.5.1.2 | 0.71 | 0.83 | 0.16 | 0.0061 | glutaminase |
| 56 | EC 4.2.1.12 | 0.23 | 0.29 | 0.17 | 0.0040 | phosphogluconate dehydratase (Edd) |
| 62 | EC 3.5.4.25 | 0.42 | 0.52 | 0.19 | NA | GTP cyclohydrolase II |
| 75 | EC 3.1.3.97 | 0.2 | 0.14 | 0.31 | 0.0029 | 3',5'-nucleoside bisphosphate phosphatase |
| 80 | EC 4.1.1.3 | 0.24 | 0.33 | 0.12 | 0.0047 | oxaloacetate decarboxylase |
| 80 | EC 4.1.1.3 | 0.24 | 0.33 | 0.12 | 0.0047 | oxaloacetate decarboxylase (Na+ extruding) |
| 82 | EC 1.1.1.86 | 0.74 | 0.66 | 0.13 | 0.0058 | ketol-acid reductoisomerase (NADP+) |
| 87 | EC 1.2.1.3 | 1.56 | 1.76 | 0.17 | NA | aldehyde dehydrogenase (NAD+) |
| 116 | EC 6.3.2.6 | 0.81 | 0.74 | 0.17 | NA | phosphoribosylaminoimidazolesuccinocarboxamide synthase |
| 118 | EC 4.2.1.3 | 1.07 | 1.26 | 0.11 | 0.0098 | aconitate hydratase |
| 143 | EC 1.6.5.11 | 6.05 | 4.94 | 0.12 | NA | NADH dehydrogenase (quinone) |
| 156 | EC 2.1.1.177 | 0.55 | 0.46 | 0.12 | 0.0041 | 23S rRNA (pseudouridine1915-N3)-methyltransferase |
| 161 | EC 6.2.1.5 | 1.49 | 1.23 | 0.12 | NA | succinate—CoA ligase (ADP-forming) |
| 1 | PF00316 | 0.92 | 0.55 | 0.3 | 0.0174 | Fructose-1-6-bisphosphatase, N-terminal domain |
| 3 | PF17866 | 0.77 | 0.4 | 0.21 | 0.0226 | AAA lid domain ("AAA_lid_6"; CbbX) |
| 6 | PF08406 | 0.91 | 0.61 | 0.21 | 0.0062 | CbbQ/NirQ/NorQ C-terminal |
| 8 | PF01116 | 1.18 | 0.81 | 0.17 | 0.0106 | Fructose-bisphosphate aldolase class-II |
| 9 | PF02823 | 0.75 | 0.54 | 0.19 | 0.0033 | ATP synthase, Delta/Epsilon chain, beta-sandwich domain |
| 11 | PF00834 | 1.03 | 0.87 | 0.16 | 0.0039 | Ribulose-phosphate 3 epimerase family |
| 12 | PF12774 | 0.16 | 0.04 | 0.17 | 0.0049 | Hydrolytic ATP binding site of dynein motor region |
| 13 | PF11684 | 0.29 | 0.19 | 0.22 | 0.0022 | Protein of unknown function (DUF3280) |
| 14 | PF02347 | 1.22 | 1.05 | 0.18 | 0.0026 | Glycine cleavage system P-protein |
| 15 | PF01112 | 0.27 | 0.15 | 0.24 | 0.0014 | Asparaginase |
| 16 | PF01794 | 0.64 | 0.83 | 0.16 | 0.0024 | Ferric reductase like transmembrane component |
| 17 | PF03441 | 0.77 | 0.95 | 0.18 | 0.0022 | FAD binding domain of DNA photolyase |
| 18 | PF13420 | 2.13 | 2.53 | 0.15 | 0.0035 | Acetyltransferase (GNAT) domain |
| 19 | PF03924 | 0.78 | 1.19 | 0.14 | 0.0036 | CHASE domain |
| 20 | PF14691 | 1.49 | 1.34 | 0.21 | 0.0043 | Dihydroprymidine dehydrogenase domain II, 4Fe-4S cluster |
| 21 | PF06580 | 0.47 | 0.61 | 0.18 | 0.0022 | Histidine kinase |
| 22 | PF03595 | 0.53 | 0.38 | 0.19 | NA | Voltage-dependent anion channel |
| 23 | PF03200 | 0.32 | 0.17 | 0.19 | NA | Glycosyl hydrolase family 63 C-terminal domain |
| 24 | PF03881 | 0.41 | 0.29 | 0.17 | NA | Fructosamine kinase |
| 25 | PF04172 | 0.32 | 0.48 | 0.18 | NA | LrgB-like family |

This table shows the top 20 ECs and Pfams, with lower ranked genes listed in S5 Dataset. Columns contain consensus rank (Rank), EC number or Pfam ID (Gene), average copy number in CBB-positive (CBB⁺) and CBB-negative (CBB⁻) genomes, which was probed by the enrichment analysis, average weighted ACE correlation value (ACE r; see Materials and methods), random forest importance (Imp.), and a description provided by the KEGG or Pfam databases (Description). Missing importance values (NA) indicate that those genes were filtered out in the random forest analysis.

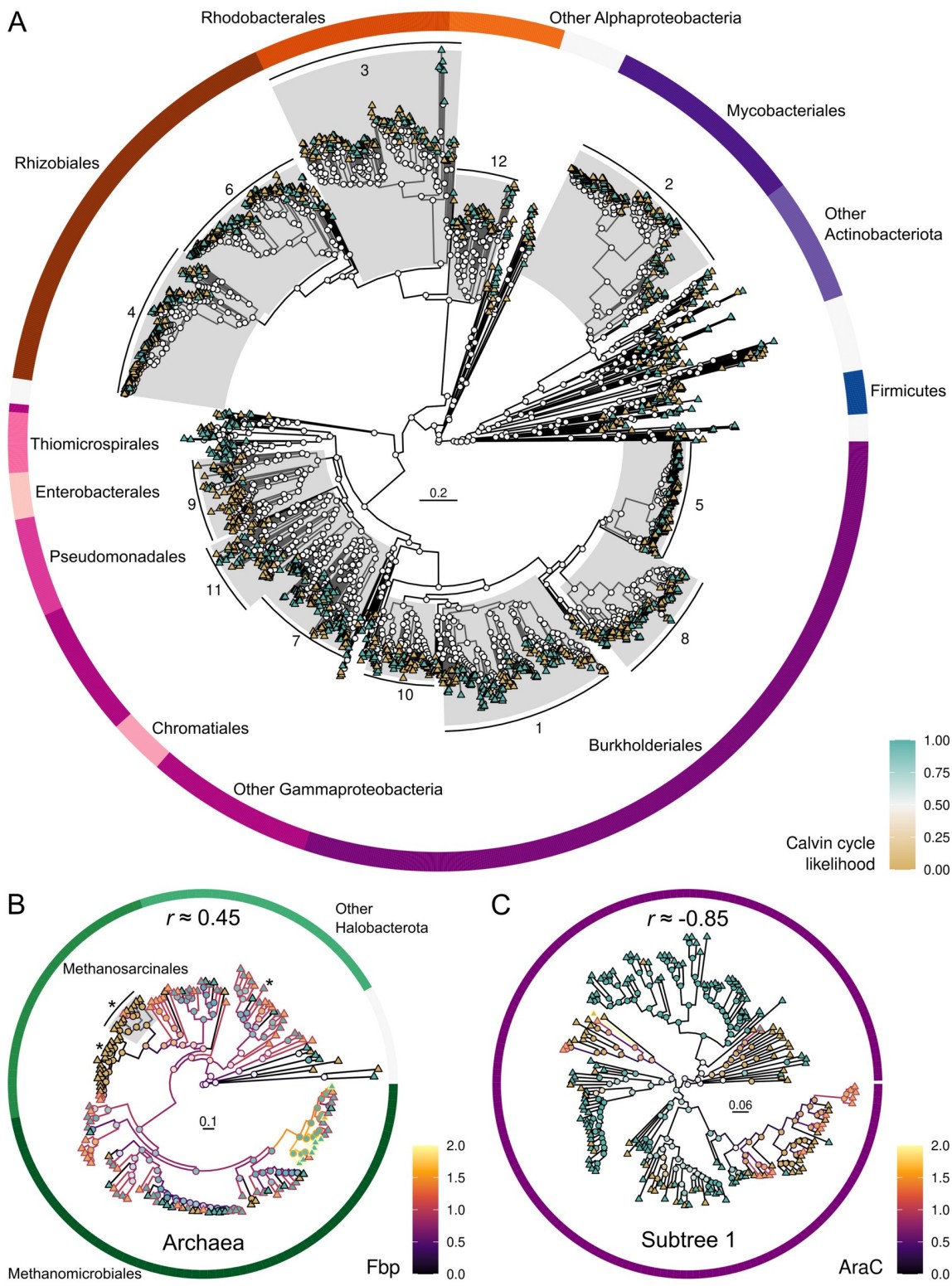

**Fig 4. Tracing the evolution of Calvin cycle genome integration.** The panels show likelihood of ancestral Calvin cycle presence (node fill color; cyan indicates CBB-positive and brown indicates CBB-negative) in bacterial subtrees (A), positive correlation (Spearman $r \approx$ 0.45) to ancestral gene copy numbers (line color) of fructose-1,6-bisphosphatase (Fbp; EC 3.1.3.11) in Archaea (B), and strong negative correlation (Spearman $r \approx$ -0.85) to ancestral gene copy numbers (line color) of transcriptional regulator AraC (PF06719) in bacterial subtree 1 (C). Each leaf node (triangles) is one contemporary genome. Outer rings indicate genome taxonomic association. Scale bars

show substitutions per site. Asterisks (*) indicate archaeal genomes encoding Rubisco activase CbbQ (PF08406). CbbQ is negatively correlated to the Calvin cycle in Archaea ($r \approx$ -0.62), which is explained by the fact that most of the archaeal genomes with CbbQ are CBB-negative (brown).

integration with central carbon metabolism (Fig 5). Several underlying patterns explain the excellent consensus ranking; Archaea was one of three subtrees that displayed positive correlation between the Calvin cycle and fructose-1,6-bisphosphatase (Fig 4B). Fructose-1,6-bisphosphatase, aldolase, and transketolase genes often located adjacently to Rubisco and Prk (Fig 3, S6 Dataset), indicating a wide-spread operon structure that has been reported previously in Calvin cycle genes [52]. An obvious explanation for this operon structure is transmission together, as illustrated by close association on megaplasmids in *Ralstonia eutropha* [37] and *Oligotropha carboxidovorans* [38], though operons can form independently of HGT to facilitate complex regulation [53]. The increased copy numbers of fructose-1,6-bisphosphatase, aldolase, and transketolase in CBB-positive genomes reflects the responsibility of these enzymes to carry out reverse fluxes compared to heterotrophic growth on glycolytic carbon sources such as glucose. Multiple copies could enable parallel evolution of Calvin cycle and glycolytic adaptations, as new copies can evolve expression levels and kinetic constants needed to retain Calvin cycle flux. Stable operation of the Calvin cycle has been proposed to depend specifically on the saturation states of fructose-1,6-bisphosphatase, aldolase, and transketolase [54].

The 3-phosphoglycerate node is important for Calvin cycle stability [31,32]. Phosphoglycerate kinase (Pgk; EC 2.7.2.3) diverts fixed carbon to RuBP regeneration and gluconeogenesis (3PG to BPG in Fig 5) and was enriched in CBB-positive genomes (0.88 copies compared to 0.77), yielding a consensus rank of 939. Phosphoglycerate mutase (Pgm) competes with Pgk to divert 3-phosphoglycerate to the TCA cycle for biomass synthesis (3PG to 2PG in Fig 5). Pgm isozymes have been shown to regulate Calvin cycle and glycolysis flux in response to changes in growth conditions [55]. We observed enrichment for 2,3-bisphosphoglycerate-dependent Pgm (EC 5.4.2.11) in CBB-positive genomes (0.49 copies compared to 0.36; consensus rank 563), but not for 2,3-bisphosphoglycerate-independent Pgm (EC 5.4.2.12; 0.40 copies compared to 0.43; consensus rank 5,249). Recently, post-translational inhibition of Pgm, carried out by the protein PirC, was shown to be critical for glycogen formation in Cyanobacteria [56]. In our dataset, PirC (PF08865) was found in only one CBB-negative *Bradyrhizobium* ORF, consistent with being a Cyanobacteria-specific adaptation.

Ribulose-phosphate 3-epimerase (EC 5.1.3.1; Xu5P to Ru5P in Fig 5) and ribose-5-phosphate isomerase (EC 5.3.1.6; R5P to Ru5P in Fig 5) held consensus ranks 316 (11 for the N-terminus PF00834) and 170. Although occasionally observed adjacent to Rubisco or Prk, the median distance for ribulose-phosphate 3-epimerase was five genes, and more than 280 genes for ribose-5-phosphate isomerase (S6 Dataset). The good ranking of these reactions can be attributed to their immediate connection to the Prk substrate ribulose-5-phosphate.

Our analysis generated poor consensus ranks for the three remaining core enzymes of the Calvin cycle, *i.e.* glyceraldehyde-3-phosphate dehydrogenase (EC 1.2.1.12 and 1.2.1.59), transaldolase (EC 2.2.1.2), and triose phosphate isomerase (EC 5.3.1.1), which ranked 1840[th] (triose phosphate isomerase) or worse. Glyceraldehyde-3-phosphate dehydrogenase and triose phosphate isomerase showed average gene copy numbers close to one (0.8–0.9) regardless of CBB status, emphasizing their contribution to glycolysis and gluconeogenesis in all organisms.

Directly connected to the central Calvin cycle metabolites, are the Entner-Doudoroff (ED) and oxidative pentose phosphate (OPP) pathways. Glycogen usage via the ED and OPP pathways could restore depleted Calvin cycle intermediates, particularly during transition between

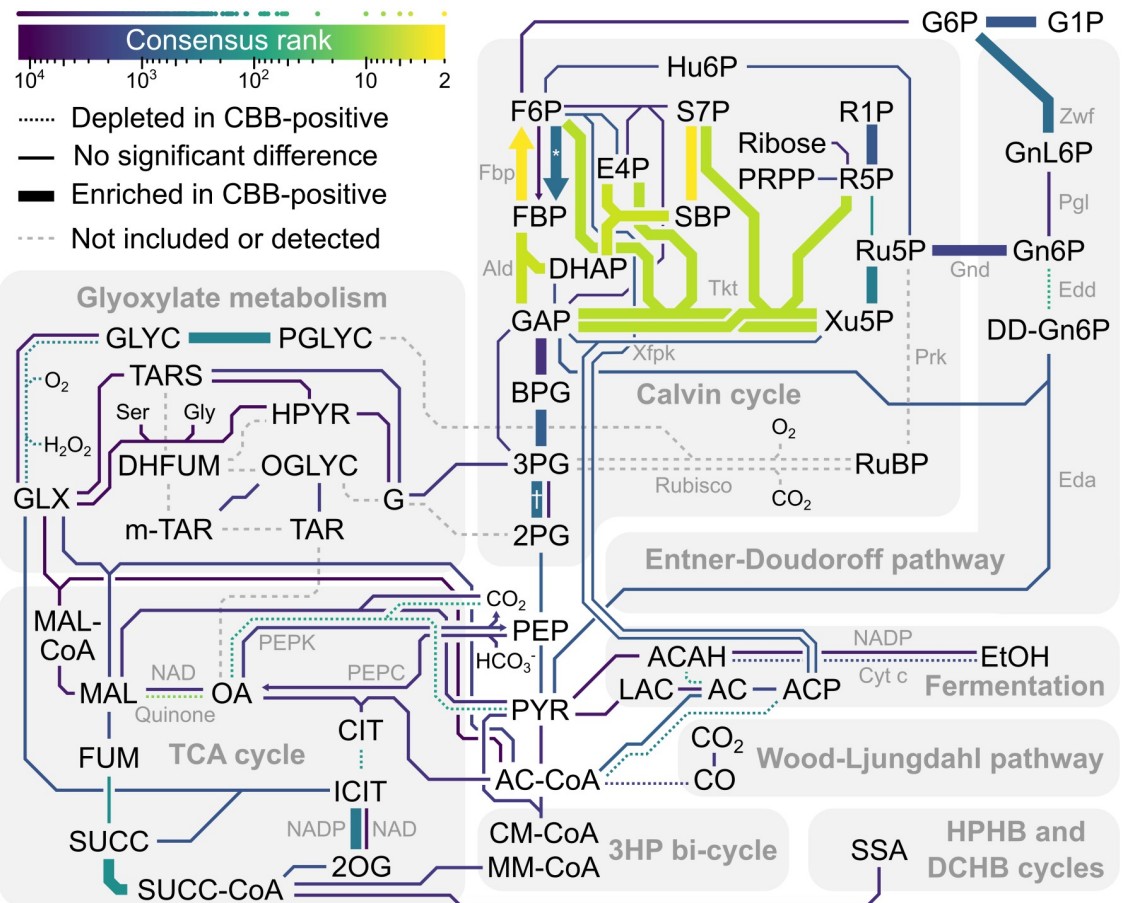

**Fig 5. Central carbon metabolism and the pentose phosphate pathway represent a hotspot for Calvin cycle adaptations.** Color indicates the consensus rank of enzymes on a logarithmic scale. Points above the color scale bar represent the consensus rank of individual enzymes. Line thickness indicates whether the enzyme-encoding genes were enriched or depleted in CBB-positive genomes. Dashed lines indicate that the enzyme was not detected or that it was removed because it was Prk or Rubisco (see Materials and methods). Co-factors and small molecules such as $CO_2$ have been omitted from most reactions. Arrows are used where enzymes that mainly catalyze specific directions rank differently. Special characters indicate pyrophosphate-dependent phosphofructo-1-kinase (*) and 2,3-bisphosphoglycerate-dependent phosphoglycerate mutase (†). The map is based on relevant subsystems of KEGG's central carbon metabolism map (*map01200*) and related maps. The logarithm of consensus ranks for Enzyme Commission (EC) numbers were normalized to the range 0 to 1 and encoded as color. We also encoded significant EC enrichment or depletion in CBB-positive genomes as different colors. The EC-to-color tables were submitted to KEGG's pathway mapping tool (https://www.genome.jp/kegg/tool/map_pathway2.html) to yield annotated maps that were then used as templates for drawing the figure. Note that the reaction SBP to S7P is represented by EC 3.1.3.11, rather than the eukaryotic SBPase EC 3.1.3.37, assuming that EC 3.1.3.11 represents bifunctional F/SBPase (Fbp). Also note that when multiple ECs mapped to the same reaction, only the best ranking EC color was used, unless special patterns of interest were present, *e.g.* 3PG to 2PG (†). Abbreviations: 2OG, 2-oxoglutarate; 2PG, 2-phosphoglycerate; 3HP, 3-hydroxypropionate; 3PG, 3-phosphoglycerate; AC, acetate; ACAH, acetaldehyde; AC-CoA, acetyl-CoA; ACP, acetyl phosphate; Ald, fructose-bisphosphate aldolase; BPG, 1,3-bisphosphoglycerate; CIT, citrate; CM-CoA, citramalyl-CoA; Cyt c, cytochrome c; DCHB, dicarboxylate-hydroxybutyrate; DD-Gn6P, 2-dehydro-3-deoxy-gluconate-6-phosphate; DHAP, dihydroxyacetone phosphate; DHFUM, dihydroxyfumarate; E4P, erythrose-4-phosphate; Eda, 2-dehydro-3-deoxy-phosphogluconate aldolase; Edd, 6-phosphogluconate dehydratase; EtOH, ethanol; F6P, fructose-6-phosphate; FBP, fructose-1,6-bisphosphate; Fbp, fructose 1,6-bisphosphate phosphatase; FUM, fumarate; G1P, glucose-1-phosphate; G6P, glucose-6-phosphate; G, glycerate; GAP, glyceraldehyde-3-phosphate; GLX, glyoxylate; Gly, glycine; GLYC, glycolate; Gnd, 6-phosphogluconate dehydrogenase; Gn6P, gluconate-6-phosphate; GnL6P, glucono-1,5-lactone 6-phosphate; HPHB, hydroxypropionate-hydroxybutyrate; HPYR, hydroxypyruvate; Hu6P, arabino-3-hexulose-6-phosphate; ICIT, isocitrate; LAC, lactate; MAL, malate; MAL-CoA, malyl-CoA; MM-CoA, methylmalonyl-CoA; m-TAR, meso-tartrate; OA, oxaloacetate; OGLYC, oxaloglycolate; PEP, phosphoenolpyruvate; PEPC, phosphoenolpyruvate carboxylase; PEPK, phosphoenolpyruvate carboxykinase; Pgl, 6-phosphogluconolactonase; PGLYC, phosphoglycolate; PRPP, 5-phosphoribosyl 1-pyrophosphate; PYR, pyruvate; R1P, ribose-1-phosphate; R5P, ribose-5-phosphate; Ru5P, ribulose-5-phosphate; RuBP, ribulose-1,5-bisphosphate; S7P, sedoheptulose-7-phosphate; SBP, sedoheptulose-1,7-bisphosphate; Ser, serine; SSA, succinate semialdehyde; SUCC, succinate; SUCC-CoA, succinyl-CoA; TAR, tartrate; TARS, tartronate semialdehyde; TCA, tri-carboxylic acid; Tkt, transketolase; Xfpk, phosphoketolase; Xu5P, xylulose-5-phosphate; Zwf, glucose-6-phosphate dehydrogenase.

growth conditions [57]. The ED and OPP pathways transform glucose 6-phosphate into ribulose-5-phosphate or glyceraldehyde-3-phosphate and pyruvate (Fig 5). The most prominent ED and OPP enzymes in our analysis were Edd (EC 4.2.1.12) at consensus rank 56, Gnd at consensus rank 100 (PF00393), and Zwf at consensus rank 579 (NADP-dependent; EC 1.1.1.49). Edd was slightly depleted in CBB-positive genomes (0.23 copies compared to 0.29) and negatively correlated in subtrees 1 and 9. Meanwhile, CBB-positive genomes were slightly enriched in NADP-dependent Zwf (0.84 copies compared to 0.71) and Gnd (0.86 copies compared to 0.65 for PF00393, 0.33 versus 0.25 for NADP-dependent EC 1.1.1.44, and 0.051 versus 0.023 for NAD-dependent EC 1.1.1.343). Furthermore, NAD-dependent Gnd was positively correlated in subtree 1. Thus, CBB-positive Bacteria and Archaea (non-cyanobacterial) may benefit from using Zwf (G6P to GnL6P in Fig 5) and Gnd (Gn6P to Ru5P in Fig 5) to restore ribulose-5-phosphate from stored glycogen. Judging by the depletion of Edd and close to poor ranking of Eda (1,145th), the CBB-positive microbes utilize the OPP shunt. The OPP shunt is less carbon efficient than the full ED shunt, but provides additional NADPH through Gnd, which may be useful in a scenario limited by energy rather than $CO_2$ availability.

Phosphoketolase (Xfpk in Fig 5) is an alternative route from the Calvin cycle intermediates xylulose-5-phosphate and fructose-6-phosphate to acetyl-CoA, via acetyl phosphate, that prevents the decarboxylation of pyruvate. On average poor ranking of Xfpk (1,074 for EC 4.1.2.9, and 8,215 for EC 4.1.2.22) and low copy numbers in both CBB-positive and CBB-negative microbes (0.11 versus 0.07 for EC 4.1.2.9, and 0.20 versus 0.18 for EC 4.1.2.22; not significant) indicates that the phosphoketolase pathway is not a widely embraced adaptation to the Calvin cycle.

## The Calvin cycle is accompanied by increased carbon storage capacity

Once carbon has been fixed by a stably operating Calvin cycle, it must be distributed to growing biomass or saved for later use. The top 200 genes, by consensus rank, included twelve sugar metabolism and carbon storage proteins (EC 2.4.1.1, PF00343, PF03200, PF17167, PF03065, PF03881, EC 2.7.7.27, PF00953, PF06165, PF09492, PF05116, and PF10091), *e.g.* carbohydrate/glycogen phosphorylase, glycosyl hydrolase, and pectic acid lyase, that were all enriched in CBB-positive genomes. When energy is readily available, CBB-positive organisms stockpile sugar in order to survive nutrient or energy limitation, as a sink for electron overflow [58], or as a source for supplying Calvin cycle intermediates [57]. Polyhydroxybutyrate (PHB), a well-studied carbon storage polymer [59], was not a general Calvin cycle adaptation. Instead, PHB depolymerase (PF10503; consensus rank 1,244) had a negative correlation in subtree 9, and the PHB accumulation regulator (PF05233 and PF07879) was significantly depleted in CBB-positive genomes.

## Carbon concentrating and recycling mechanisms enhance Calvin cycle operation

Adaptations to the Calvin cycle may go beyond enzymes in central carbon metabolism. For example, the carboxysome is a microcompartment made of protein that houses Rubisco and carbonic anhydrase, ensuring high concentration of $CO_2$ for optimal $CO_2$ fixation by Rubisco [60]. Subtree 1, dominated by Burkholderiales, showed positive correlations between the Calvin cycle and the carboxysome shell protein CsoS as well as the carboxysome-related protein ethanolamine utilization protein EutN (PF03319). Furthermore, the bacterial microcompartment protein BMC (PF00936) was enriched in CBB-positive genomes (0.27 copies per genome compared to 0.13). While CsoS (PF12288) occupied consensus rank 824, with 0.12 copies per CBB-positive genome (0.02 in CBB-negative), the associated carbonic anhydrase (PF08936)

ranked 263th, with 0.16 copies per genome (0.03 in CBB-negative), suggesting that the carbonic anhydrase might operate independently of the shell protein, as shown for *Ralstonia eutropha* [61]. Optimal Rubisco function also requires activase chaperones [62,63], *i.e.* CbbQ (PF08406) and CbbX (PF17866), which occupied top consensus ranks and were genomically associated with Rubisco (Fig 3). A lack of accessory proteins might make the introduction of Rubisco into new hosts difficult, but it was not necessary in *E. coli* [29,30]. Among Archaea, CbbQ genes were primarily restricted to the CBB-negative Methanosarcinales clade (asterisks in Fig 4B), which harbors *e.g. Methanococcoides* that uses Rubisco for nucleoside metabolism [64]. CBB-positive Archaea may use a different activase, or may not need one. For example, the *Synechococcus* sp. PCC 7942 Rubisco is independent of RbcX activase [65] with RbcX supporting carboxysome assembly [66]. Consistent with excluding cyanobacterial genomes, we did not detect RbcX (PF02341).

Photorespiration, *i.e.* when Rubisco fixes $O_2$ instead of $CO_2$, produces 2-phosphoglycolate, which is subsequently oxidized to glyoxylate by phosphoglycolate phosphatase (EC 3.1.3.18; PGLYC to GLYC in Fig 5) and glycolate oxidase (EC 1.1.3.15; GLYC to GLX in Fig 5). These two enzymes ranked 255th and 268th, and while the former was enriched in CBB-positive organisms (0.81 copies compared to 0.68), the latter was rare and depleted (0.07 copies compared to 0.1). Cyanobacteria, which were excluded in this analysis, benefit from having more than one copy of phosphoglycolate phosphatase to prevent inhibition of Calvin cycle enzymes caused by 2-phosphoglycolate [67]. However, other CBB-positive microbes do not seem to have a rich complement of phosphoglycolate phosphatase in general (0.81 copies per genome). Looking at CBB-positive genomes individually, 416 had no copies, 447 had one copy, 143 had two copies, 13 had three, and one genome (*Enterovibrio calviensis*) had four copies. Out of these 157 genomes with more than one copy, 118 belonged to Gammaproteobacteria and 37 belonged to Alphaproteobacteria, *i.e.* a total of 99%, while these groups constitute 73% of the analyzed genomes (Fig 2). The microbes with putative phosphoglycolate phosphatase isozymes belonged mainly to the orders Burkholderiales (62 genomes), Rhizobiales (23), Rhodobacterales (9), Thiomicrospirales (9), Chromatiales (8) and Enterobacterales (8). These findings suggest that some CBB-positive microbes may benefit from having phosphoglycolate phosphatase isozymes like observed in Cyanobacteria. Glyoxylate carbons can be recycled or eliminated for example by the glycerate pathway in *Ralstonia eutropha*, the malate cycle, or the photorespiratory C2 cycle [68,69]. Tartronate semialdehyde reductase (EC 1.1.1.60; TARS to G in Fig 5) and tartrate dehydrogenase (EC 1.1.1.93; OGLYC to TAR/m-TAR in Fig 5) represent the glycerate pathway, but these enzymes showed no enrichment, and only the former was significant in the ACE analysis, reporting both positive (subtree 5) and negative correlation (subtree 4). The malate cycle (EC 2.3.3.9, EC 1.1.1.38–40, EC 1.2.4.1, 1.8.1.4, and 2.3.1.12) did not rank better than 2,246th and was therefore not a Calvin cycle-specific adaptation, although gene copy numbers of 0.19–1.4 would support operation in certain organisms. Photorespiration in non-oxygenic autotrophs may be less important than in oxygenic photosynthetic organisms such as Cyanobacteria. Another explanation could be use of the Calvin cycle to increase carbon yield on sugar rather than an exclusively autotrophic lifestyle [70], which could lower the selection pressure for evolving extensive photorespiration routes. Nevertheless, another 2-phosphoglycolate "salvage" alternative is the glyoxylate shunt operated by isocitrate lyase, which generates isocitrate from succinate and glyoxylate (SUCC and GLX to ICIT in Fig 5). Lower aconitase (EC 4.2.1.3) and higher NADP-dependent isocitrate dehydrogenase (EC 1.1.1.42) rates could facilitate flux along the glyoxylate shunt towards 2-oxoglutarate (CIT to ICIT to 2OG in Fig 5), releasing $CO_2$ for re-fixation in the Calvin cycle. Aconitase was indeed depleted in CBB-positive genomes (1.1 copies compared to 1.3), while isocitrate dehydrogenase was enriched (0.52 copies compared to 0.44).

## The Calvin cycle is accompanied by adaptations for energy acquisition

Autotrophic metabolism is fueled by inorganic energy sources such as light (photosynthesis), iron [71], sulfur/sulfide [72,73], or molecular hydrogen [74]. Based on the enrichment of energy metabolism among essential genes in Cyanobacteria [75,76], we expected an association between energy metabolism and the non-cyanobacterial Calvin cycle. Indeed, *e.g.* sulfur oxidation protein SoxZ (PF08770) from autotrophic sulfur-oxidizing bacteria, and six hydrogenase domains (PF01155, PF01924, PF17788, PF07503, PF00374, and PF02769) showed higher gene copy numbers in CBB-positive genomes. The best ranking hydrogenase gene occupied consensus rank 34 (PF01155). Each hydrogenase gene co-located with Rubisco or Prk on a plasmid at most on three occasions, and located > 80 genes away in general, suggesting that transmission through plasmid or adjacency is rare (S6 Dataset). However, note that only 163 (16%) of the 1,020 CBB-positive genomes had contigs confirmed to be chromosomes or plasmids. Nevertheless, we interpret the good ranking of hydrogenase as either adaptations to the Calvin cycle, or as a genetic background that favored the acquisition of the Calvin cycle.

Ammonia monooxygenase (EC 1.14.99.39) had on average 0.11 copies per CBB-positive genome, but was nearly absent from CBB-negative genomes. The enzyme held one of the highest importance values in the random forest analysis, and showed significant correlation to the Calvin cycle in subtree 1. Despite this, the consensus rank was only 2,322. Ammonia monooxygenase catalyzes the first step of nitrification, supplying energy for *e.g.* $CO_2$ fixation. Ammonia oxidizing Archaea use the 3-hydroxypropionate/4-hydroxybutyrate cycle for autotrophic growth [77], while certain ammonia oxidizing Bacteria, *e.g. Nitrosospira*, *Nitrosococcus*, and *Nitrosomonas*, use the Calvin cycle [78]. The poor ranking, but near absence in CBB-negative genomes, indicates that ammonia monooxygenase may be a niche adaptation to the Calvin cycle.

CBB-positive genomes showed increased gene copy numbers of electron transport chain components such as ATP synthase (PF02823; Fig 3, Table 1), proton-conducting membrane transporter (PF00361), cytochrome C7 (PF14522), and NADH-ubiquinone oxidoreductase (PF01059), among the consensus top 50 genes. These genes may reflect an energy management unique to autotrophs.

Based on the photosynthetic bacterium *Bradyrhizobium* sp. ORS 278, we considered organisms photosynthetic that carried at least three photosynthetic reaction center Pfams (S2 Table). Since 115 CBB-positive genomes and 113 CBB-negative genomes encoded photosynthesis capability, photosynthesis was not specific for the Calvin cycle.

## Metabolic and genetic regulation associated with the Calvin cycle

Calvin cycle function in a new host is likely to require evolution of control at different levels, such as regulation of enzyme activities and gene transcription. For example, when the Calvin cycle was established in *E. coli*, its stable operation required mutation of both the PEP synthetase regulator *ppsR* and the master metabolic transcription regulator *crp* [31]. With the ACE analysis, we found evidence of several transcription factors that were selected for or against in specific phylogenetic subtrees. The AraC-type transcriptional regulator (PF06719), showed one of the strongest negative correlations, with $r \approx -0.85$ in subtree 1 (Fig 4C), yet its poor consensus rank (2,117) suggests it is not a general adaptation across multiple subtrees. A helix-turn-helix motif associated with AraC proteins (PF00165) had a good consensus rank of 234, due to its depletion in CBB-positive genomes (13 copies compared to 16), and negative correlation in subtree 1 and in Archaea. The arabinose binding domain of AraC (PF12625) was even more prominent at consensus rank 52, and showed significant depletion in CBB-positive genomes (1.4 copies compared to 2.0). AraC senses arabinose and regulates the arabinose

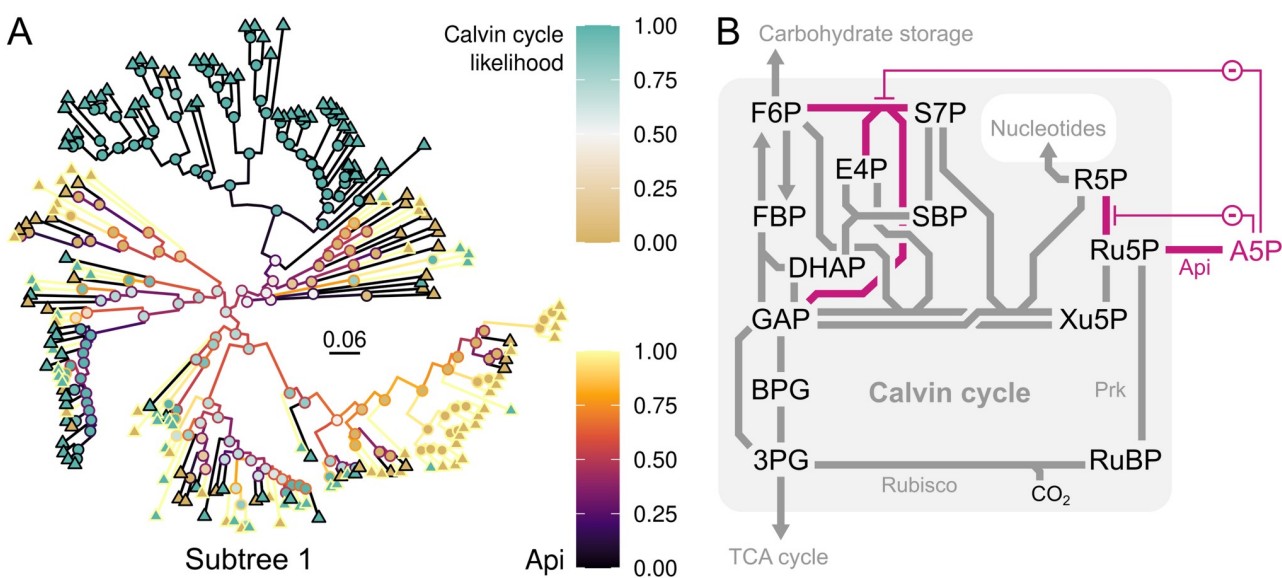

**Fig 6. Calvin cycle-positive organisms avoid metabolite-level regulation that may disturb cycle function.** The enzyme arabinose-5-phosphate isomerase (Api; EC 5.3.1.13) was negatively correlated with the Calvin cycle (Spearman $r \approx$ -0.61) in subtree 1 (A), illustrated by likelihood of ancestral Calvin cycle presence (node fill color) and ancestral Api gene copy numbers (line color). The scale bar (A) shows substitutions per site. Api interferes with Calvin cycle operation (B) by converting ribulose-5-phosphate to arabinose-5-phosphate (A5P). A5P inhibits (-) transaldolase (GAP and S7P to F6P and E4P) and ribose-5-phosphate isomerase (R5P to Ru5P). The map is based on KEGG's central carbon metabolism map (*map01200*). Abbreviations: 3PG, 3-phosphoglycerate; A5P, arabinose-5-phosphate; Api, arabinose-5-phosphate isomerase; BPG, 1,3-bisphosphoglycerate; DHAP, dihydroxyacetone phosphate; E4P, erythrose-4-phosphate; F6P, fructose-6-phosphate; FBP, fructose-1,6-bisphosphate; GAP, glyceraldehyde-3-phosphate; R5P, ribose-5-phosphate; Ru5P, ribulose-5-phosphate; RuBP, ribulose-1,5-bisphosphate; S7P, sedoheptulose-7-phosphate; SBP, sedoheptulose-1,7-bisphosphate; TCA, tri-carboxylic acid; Xu5P, xylulose-5-phosphate.

operan [79]. AraC's negative correlation in subtree 1 was matched by arabinose-5-phosphate isomerase (EC 5.3.1.13; Fig 6A). CBB-positive microbes may have a smaller set of AraC-type regulators than CBB-negative because of not utilizing the pentose arabinose, or because they must avoid arabinose metabolism enzymes. Arabinose-5-phosphate isomerase connects to the Calvin cycle by operating on ribulose-5-phosphate (Ru5P in Figs 5 and 6B). Diverting flux near ribulose-5-phosphate may have a significant effect on Calvin cycle operation since this metabolite is the substrate of Prk. Furthermore, arabinose-5-phosphate inhibits ribose-5-phosphate isomerase [80] and transaldolase [81], demonstrated for enzymes from *E. coli* and *Francisella tularensis*, respectively, which may disturb Calvin cycle operation (Fig 6B).

## Many top ranked genetic adaptations have unknown functions

Pfam domains of unknown function (DUFs) and uncharacterized protein families (UPFs) may play previously unknown roles in adaptation to the Calvin cycle. We found 20 DUFs/UPFs among the top 200 Pfams, that is 10%, while they account for 22.6% of the full Pfam 32.0. Genes of unknown function were less frequent also among essential genes of the photoautotroph *Synechoccocus elongatus* PCC 7942, which was attributed to essential gene conservation and research focusing on phenotypes that can be measured [75]. DUF4156 (PF13698) was the most enriched DUF/UPF (0.14 copies compared to 0.031 in CBB-negative genomes), and held consensus rank 652. DUF2958 (PF11171) and DUF333 (PF03891) achieved strong absolute correlations in the ACE analysis, but both showed positive and negative correlations in different subtrees, and ranked poorly at consensus ranks 1,429 and 3,151. Finally, DUF3280 (PF11684), may be of special interest as it occupied the top 20 Pfams by consensus rank (Table 1).

## Conclusion

Our comparison of genomes carrying the Calvin cycle and their closest relatives lacking this characteristic identified a wide range of adjustments to central carbon metabolism, sugar metabolism, and energy metabolism supporting an autotrophic lifestyle. Some adaptations were general, such that their enrichment or depletion were observed in multiple phylogenetic subtrees, such as the closely associated enzymes fructose-1,6-bisphosphatase, aldolase, and transketolase, while others were specialized, such as ammonia monooxygenase. The importance of CBB-adjacent reactions, such as that catalyzed by arabinose-5-phosphate isomerase, that may alter concentrations of cycle intermediates or produce Calvin cycle inhibitors, may become clear with more biochemical detail of Calvin cycle enzymes, or interpretation with models that can detect metabolic effects on cycle stability [54]. Adaptations also differ depending on the local evolutionary context; The Rubisco activase CbbQ was either positively or negatively correlated to the Calvin cycle, depending on what subtree was queried in the ACE analysis. The domains of unknown function were similarly diverse in their responses to the presence of the Calvin cycle. Importantly, many adaptations were not due to co-transmission with the Calvin cycle, as demonstrated by significant depletions, or by occupation of distant genomic loci. To conclude, we suggest that future metabolic engineering projects should learn from adaptations in the host organism's closest relatives to reach the most efficient nature-aided design. Future analyses may also guide metabolic and protein engineering for improved autotrophic traits by identifying mutations in key genes using methods similar to those presented here.

## Materials and methods

Data analysis was carried out in R v.3.6.1 with tidyverse v.1.2.1 (https://www.tidyverse.org/), and doMC v.1.3.6 and foreach v.1.4.7 for parallel computation. Random forest training was done in Python v.3.5.6. Bash commands were parallelized using GNU Parallel v.20141022 [82]. Scripts are available at https://github.com/Asplund-Samuelsson/redmagpie. Software was run on Ubuntu Linux 18.04.3 LTS (16 CPU cores and 128 GB RAM) and 20.04.1 LTS (12 CPU cores and 32 GB RAM).

24,706 archaeal and bacterial genomes listed as species representatives in GTDB release 89 [83,84] were downloaded from NCBI on 16–19 August 2019 in nucleotide FASTA format. Genome completeness CheckM [85] values were provided by GTDB. We identified ORFs with stand-alone ORFfinder v.0.4.3 (https://www.ncbi.nlm.nih.gov/orffinder/), using the bacterial, archaeal and plant plastid translation table (option -g 11), requiring ORF length ≥ 300 (option -ml 300), and excluding ORFs completely surrounded by another ORF (option -n true).

Two Hidden Markov Models (HMMs) were constructed for identification of Rubisco (EC 2.7.1.19) and Prk (EC 4.1.1.39). Amino acid sequences representing KEGG (https://www.genome.jp) orthologs K01601 (Rubisco; 917 sequences) and K00855 (Prk; 933 sequences) were downloaded from UniProt (https://www.uniprot.org/) on 25 July 2019, and clustered at 70% identity using cd-hit v.4.7 [86,87], yielding 150 and 106 cluster representatives for Rubisco and Prk. The representatives were aligned with MAFFT v.7.271 [88], followed by removal of positions with > 50% gaps, and then removal of sequences with > 50% gaps, using seqmagick v0.6.2 (https://fhcrc.github.io/seqmagick/). The remaining 144 Rubisco and 105 Prk sequences were re-aligned using MAFFT and encoded as HMMs using hmmbuild from hmmer v3.1b2 (http://hmmer.org/). The HMMs yielded bit scores ≥ 25.3 for Rubisco and ≥ 78.2 for Prk when used with hmmsearch on the sequences from UniProt, thus assuring HMM quality.

The HMMs representing Rubisco and Prk were used with hmmsearch on the archaeal and bacterial ORFs to classify genomes as Calvin cycle positive (CBB-positive) or negative (CBB-

negative). The amino acid sequences identified by hmmsearch (sequence E-value < 0.01) were submitted to KEGG BlastKOALA v2.2 (https://www.kegg.jp/blastkoala/) on 21–23 August 2019, and also subjected to DeepEC (https://bitbucket.org/kaistsystemsbiology/deepec) commit b7e4546 [48] EC number classification. Sequences that only yielded an unexpected KEGG ortholog or EC were removed. To eliminate form IV Rubisco, which lacks carboxylase activity, the Rubisco sequences were aligned using MAFFT, and then filtered to sequences possessing the critical catalytic lysine residue corresponding to position 174 in the alignment by Hanson and Tabita [89]. Filtered Rubisco sequences and 181 examples of form I-III Rubisco and form IV Rubisco-like proteins (RLPs) identified by Tabita and colleagues [19] were aligned using MAFFT and used for tree construction with FastTreeMP v.2.1.8 SSE3 [90] to confirm removal of RLPs (S1 Fig). Genomes with both Rubisco and Prk were classified as CBB-positive, but cyanobacterial genomes were excluded from the downstream analysis.

Distances between CBB-positive genomes and all other genomes (CBB-negative) were calculated from the GTDB archaeal and bacterial core protein alignments using FastTreeMP and the option -*makematrix*. Pairs of CBB-positive and CBB-negative genomes with the shortest distance were selected by looping, excluding already selected genomes, until all CBB-positive genomes had been selected. We thereby obtained a dataset with the same number of CBB-positive and CBB-negative example genomes, and with the closest possible similarity between CBB-negative and CBB-positive genomes.

The example genome ORFs were annotated with DeepEC and Pfam (https://pfam.xfam.org/) release 32.0 [91] using hmmsearch and the trusted HMM cutoffs. Rubisco (EC 2.7.1.19, PF02788, PF00016, and PF00101) and Prk (EC 4.1.1.39, and PF00485) are expected in CBB-positive genomes given our definition and were therefore excluded.

Example genomes that were at least 95% complete were subjected to an enrichment analysis in R comparing the count of each EC or Pfam between CBB-positive and CBB-negative genomes using a Wilcoxon rank sum test (function *wilcox.test*). A Benjamini-Hochberg *q* value < 0.05 (function *p.adjust*) was considered significant.

For the ACE analysis, we generated phylogenetic trees of Archaea and Bacteria using the GTDB core protein alignments and FastTreeMP (approximately maximum likelihood based on the JTT+CAT model). Using phytools v.0.7–47 [92] for R, trees were midpoint-rooted (function *midpoint.root2* from MidpointRooter v.0.1.0; https://github.com/bwemheu/MidpointRooter) and pruned to the example genomes (function *drop.tip*). Monophyletic bacterial subtrees (*B*) were selected based on four factors multiplied to yield a similarity score *s* relative to the archaeal tree (*A*), *i.e.* the ratios of number of CBB-positive genomes (*P*), CBB-negative genomes (*N*), edge length coefficient of variation (*c_v*), and maximum height (*h*), as shown in Eq 1.

$$s = 1/exp\left(\left|ln\left(\frac{P_B}{P_A}\right)\right| + \left|ln\left(\frac{N_B}{N_A}\right)\right| + \left|ln\left(\frac{c_{v,B}}{c_{v,A}}\right)\right| + \left|ln\left(\frac{h_B}{h_A}\right)\right|\right) \tag{1}$$

First, we obtained all subtrees (function *extract.clade*) with 50 to 300 taxa and calculated their similarity scores. The subtree with the highest similarity score was then selected by looping, excluding subtrees with nodes (function *getDescendants*) that overlapped with already selected subtrees, until no more subtrees could be selected. The likelihood of being CBB-positive, and the count of every EC and Pfam, were estimated for ancestral nodes using the *ace* and *fastAnc* functions, respectively. The ancestral CBB-positive likelihoods were subjected to Spearman correlation to the ancestral EC or Pfam counts (functions *cor* and *cor.test*). Additionally, ancestral nodes were classified as CBB-positive if the likelihood for that state was > 0.5, and otherwise as CBB-negative. The distribution of ancestral EC or Pfam counts

were then compared between the two classes using a Wilcoxon rank sum test (function *wilcox. test*). A Benjamini-Hochberg *q* value < 0.001 (function *p.adjust*) for both the Spearman correlation test and the Wilcoxon rank sum test was considered significant. Correlations were visualized on trees using *ggtree* [93].

A random forest classifier was implemented in Python scikit-learn v.0.23.1 (*sklearn*) using CBB-positive and CBB-negative genomes as training and testing examples, and EC or Pfam counts as features. The 600 most promising features were first selected using logistic regression with the *liblinear* solver (function *LogisticRegression* from *sklearn.linear_model*) and looped removal of the ten features with the lowest absolute feature coefficients using function *RFE* from *sklearn.feature_selection*. Random forests (function *RandomForestClassifier* from *sklearn. ensemble*) with 500 estimators were trained on a randomly sampled ¾ subset of the example genomes and tested on the remaining ¼ of the example genomes. The feature importance values from 100 random forests produced a final mean feature importance for each of the 600 selected ECs and 600 selected Pfams.

Features were given a rank (R function *rank*) based on *q* (increasing) for the enrichment, sum of absolute Spearman *r* among subtrees weighted by *q* values (decreasing) for ACE, and feature importance (decreasing) for the random forest analysis. Eq 2 describes how the feature ranking value was calculated for the ACE method:

$$\sum |r_w| = \sum_{T_0}^{T_n} |r| \cdot \frac{ln(q_{c,n} + q_c)}{ln(q_c)} \cdot \frac{ln(q_{W,n} + q_W)}{ln(q_W)}$$

(2)

where $T_n$ is the subtree, *r* is the Spearman correlation coefficient, $q_{c,n}$ is the correlation coefficient *q* value in subtree *n* and $q_c$ is the corresponding median across all subtrees and features, and $q_{W,n}$ is the Wilcoxon rank sum test *q* value in subtree *n* and $q_W$ is the corresponding median across all subtrees and features. The sum of weighted absolute *r* values $|r_w|$ over all subtrees was used to rank features. Feature types, *i.e.* EC or Pfam, were ranked together except in the random forest analysis since the importance values are only comparable within types. The enrichment, ACE, and random forest analysis feature ranks were correlated pair-wise using the Spearman method on features in common between the methods (functions *cor* and *cor. test*). A consensus rank for each feature was obtained by ranking the sum of ranks from the three methods. Features not included by a method were given a rank corresponding to the maximum rank within the method plus one.

Finally, we calculated the median distance in number of ORFs between feature-encoding ORFs and Rubisco and Prk ORFs in CBB-positive genomes, considering all contigs, including chromosomes, plasmids, and unplaced scaffolds, according to assembly information from NCBI. Contigs were considered to be linear, meaning that some ORFs were assigned greater distances than on circular DNA molecules.

## Supporting information

**S1 Fig. Phylogenetic analysis of Rubisco sequences indicates successful exclusion of Rubisco-like proteins and possible horizontal gene transfer.** The phylogenetic tree is based on Rubisco sequences identified in ORFs from Genome Taxonomy Database (GTDB) and 181 example Rubisco and Rubisco-like proteins (RLPs) identified by Tabita *et al.* [19]. Outer ring colors indicate the organism in GTDB carrying each Rubisco ORF, and inner ring colors indicate the Rubisco form. Rubisco sequences from genomes in GTDB identified as CBB-positive, *i.e.* containing Prk in addition to Rubisco, are indicated in gray in the inner ring. Note that Cyanobacteria, although carrying the Calvin cycle, were not included in the CBB-positive dataset in our analysis. The tree was rooted at the most recent common ancestor of all RLPs,

turning the RLPs into an outgroup indicated by light purple shading. The scale bar shows substitutions per site.
(PNG)

**S1 Table. Random sample of 50 CBB-positive genomes with literature references supporting or opposing Calvin cycle utilization and autotrophy.** The columns contain GTDB accession ID ('Accession'), species name from GTDB ('Species'; family name is given within parentheses if used for conclusion), a comment regarding Calvin cycle utilization and autotrophy or other acquired information ('Comment'), statement about Calvin cycle confirmation ('CBB'; 1 if likely confirmed, 0 if not likely confirmed, NA if there was no information), and references for the comment ('Ref.'). There was no information available for seven genomes. For the remaining 43 genomes, 30 (70%) appeared to be likely Calvin-cycle positive genomes, but only one (2%) genome appeared unlikely to be Calvin-cycle positive given its status as a human pathogen (*Mycolicibacterium mageritense*).
(PDF)

**S2 Table. Pfams associated with photosynthesis.** The table lists all Pfams that match the search terms "photosynthesis", "photosynthetic", or "photosystem". The columns contain Pfam feature ID ('Feature'), feature name ('Name'), and feature description ('Description'; the DESC line from the Pfam HMM database). Given the occurrence of three of these Pfams in the photosynthetic organism *Bradyrhizobium* sp. ORS 278, photosynthesis capability was assigned to organisms with at least three of these Pfams.
(PDF)

**S1 Dataset. Example genomes and annotations.** Tab-delimited text file with one row for each of 1,020 CBB-positive and 1,020 CBB-negative microbial genomes investigated in this study. The first row is a header with column titles. The columns contain GTDB accession ID ('Accession'), GTDB accession ID of closest relative with opposing CBB status selected through iteration ('Relative'; see Materials and methods), distance to closest relative ('Distance'), subtree association ('Subtree'; 0 for Archaea, 1–12 for Bacteria), CBB status ('CBB_status'; 0 for CBB-negative, 1 for CBB-positive), CheckM genome completeness provided by GTDB ('checkm_completeness'), GTDB taxonomy ('gtdb_taxonomy'), followed by columns with per-genome total counts for each of 12,703 genetic features, *i.e.* Pfam and Enzyme Commission (EC) number. The genetic feature counts represent input data for the enrichment, ancestral character estimation, and random forest analyses.
(ZIP)

**S2 Dataset. Enrichment analysis.** Results from Wilcoxon rank sum tests to determine significant differences in Pfam and Enzyme Commission (EC) number counts between CBB-positive and CBB-negative genomes (see Materials and methods). The columns contain rank based on 'q' ('Rank'), feature type ('Feature_Type'; DeepEC or Pfam), feature ID ('Feature'), feature name ('Name'), mean feature count in CBB-negative genomes ('mean_Negative'), mean feature count in CBB-positive genomes ('mean_Positive'), coefficient of variation of feature count in CBB-negative genomes ('CV_Negative'), coefficient of variation of feature count in CBB-positive genomes ('CV_Positive'), Wilcoxon rank sum test p-value ('p'), Benjamini-Hochberg adjusted p-value ('q'), feature description ('Description'; the DESC line from the Pfam HMM database, or the full list of enzyme names from KEGG for DeepEC), and the KEGG EC of the entry ('KEGG_EC'). Note that some DeepEC ECs were transferred to one or more new ECs in KEGG, as indicated by a discrepancy between 'Feature' (if 'Feature_Type' is DeepEC) and 'KEGG_EC'. A single feature can therefore be listed more than once.
(XLSX)

**S3 Dataset. Ancestral character estimation (ACE) analysis.** Results from ACE analysis in subtrees of Archaea and Bacteria comparing the evolution of Pfam and Enzyme Commission (EC) number counts to the evolution of the CBB-positive trait (see Materials and methods). The columns contain rank based on sum of absolute 'r' weighted by 'q_Correlation' and 'q_Wilcox' across subtrees ('Rank'), feature type ('Feature_Type'; DeepEC or Pfam), feature ID ('Feature'), feature name ('Name'), organism domain ('Domain'; Archaea or Bacteria), subtree number ('Subtree'; Archaea has only one subtree numbered '0'), ancestral feature count versus ancestral CBB-positive likelihood Spearman correlation r ('r'), significance ('Significant'; 1 if both q-values < 0.001, otherwise 0), p-value for the Spearman correlation ('p_Correlation'), Benjamini-Hochberg adjusted correlation p-value ('q_Correlation'), p-value for Wilcoxon rank sum test comparing feature count in CBB-positive, *i.e.* likelihood > 0.5, and CBB-negative ancestral nodes ('p_Wilcox'), Benjamini-Hochberg adjusted Wilcoxon rank sum test p-value ('q_Wilcox'), feature description ('Description'; the DESC line from the Pfam HMM database, or the full list of enzyme names from KEGG for DeepEC), and the KEGG EC of the entry ('KEGG_EC'). Note that some DeepEC ECs were transferred to one or more new ECs in KEGG, as indicated by a discrepancy between 'Feature' (if 'Feature_Type' is DeepEC) and 'KEGG_EC'. A single feature can therefore be listed more than once.
(XLSX)

**S4 Dataset. Random forest analysis.** Feature importances derived from random forest classification of CBB-positive and CBB-negative genomes based on Enzyme Commission (EC) number or Pfam counts. The columns contain rank based on 'Importance' ('Rank'), feature type ('Feature_Type'; DeepEC or Pfam), feature ID ('Feature'), feature name ('Name'), average feature importance for 100 random forests ('Importance'), coefficient of variation for feature importance based on 100 random forests ('CV_Importance'), feature description ('Description'; the DESC line from the Pfam HMM database, or the full list of enzyme names from KEGG for DeepEC), and the KEGG EC of the entry ('KEGG_EC'). Note that some DeepEC ECs were transferred to one or more new ECs in KEGG, as indicated by a discrepancy between 'Feature' (if 'Feature_Type' is DeepEC) and 'KEGG_EC'. A single feature can therefore be listed more than once. Also note that the importance and rank was calculated separately for DeepEC and Pfam feature types.
(XLSX)

**S5 Dataset. Consensus ranks for the three methods (enrichment, ACE, and random forest).** The columns contain the consensus rank ('Rank'), feature type ('Feature_Type'; DeepEC or Pfam), feature ID ('Feature'), feature name ('Name'), mean feature count in CBB-negative genomes ('mean_Negative'), mean feature count in CBB-positive genomes ('mean_Positive'), weighted average correlation coefficient coefficient in subtrees of the ACE analysis ('weighted_r'; see Materials and methods for details), average feature importance for 100 random forests ('Importance'), feature description ('Description'; the DESC line from the Pfam HMM database, or the full list of enzyme names from KEGG for DeepEC), and the KEGG EC of the entry ('KEGG_EC'). Note that some DeepEC ECs were transferred to one or more new ECs in KEGG, as indicated by a discrepancy between 'Feature' (if 'Feature_Type' is DeepEC) and 'KEGG_EC'. A single feature can therefore be listed more than once.
(XLSX)

**S6 Dataset. Proximity between Rubisco or Prk and other genetic features on DNA molecules in CBB-positive genomes.** The columns contain the consensus rank ('Rank'; see S5 Dataset), feature type ('Feature_Type'; DeepEC or Pfam), feature ID ('Feature'), feature name ('Name'), the Calvin cycle feature to which distance was measured ('cFeature'), whether the

genetic feature and Prk or Rubisco are on the same strand ('Strand'; Same or Opposite), the minimum observed distance in number of genes ('minD'), the median observed distance in number of genes ('medD'), the maximum observed distance in number of genes ('maxD'), the average distance in number of genes ('meanD'), the number of occurrences of the feature in the particular configuration described by the row ('Count'), number of occurrences on a chromosome ('locChr'), number of occurrences on a plasmid ('locPsm'), number of occurrences on an unknown DNA molecule type ('locUnk'; *e.g.* a contig of a draft genome), fraction of genes located on a plasmid ('fracPsm'; 'locPsm' divided by the sum of 'locPsm' and 'locChr'), feature description ('Description'; the DESC line from the Pfam HMM database, or the full list of enzyme names from KEGG for DeepEC), and the KEGG EC of the entry ('KEGG_EC'). Note that some DeepEC ECs were transferred to one or more new ECs in KEGG, as indicated by a discrepancy between 'Feature' (if 'Feature_Type' is DeepEC) and 'KEGG_EC'. A single feature can therefore be listed more than once. The table has been sorted by increasing median distance ('medD') followed by decreasing number of occurrences ('Count'). Only features located on the same DNA molecule as Rubisco or Prk were included, and only CBB-positive genomes were considered. All DNA molecules were considered as linear due to the many genomes in a state of unplaced scaffolds, which may place certain features further way from Rubisco and Prk compared to if the DNA molecule would have been modeled as a circle. Rubisco and Prk, representing ORFs that were identified by the specific method for Rubisco and Prk identification described in the Materials and methods, are included as features in order to show the proximity between Rubisco and Prk.
(XLSX)

## Acknowledgments

We are grateful to Anders Andersson (KTH) and Lars Arvestad (Stockholm University) for inspiring discussions and helpful suggestions. We would also like to thank Markus Janasch (KTH), Michael Jahn (KTH), and Johannes Yayo (KTH) for insightful comments.

## Author Contributions

**Conceptualization:** Johannes Asplund-Samuelsson, Elton P. Hudson.

**Data curation:** Johannes Asplund-Samuelsson.

**Formal analysis:** Johannes Asplund-Samuelsson.

**Funding acquisition:** Elton P. Hudson.

**Investigation:** Johannes Asplund-Samuelsson.

**Methodology:** Johannes Asplund-Samuelsson.

**Project administration:** Johannes Asplund-Samuelsson, Elton P. Hudson.

**Resources:** Elton P. Hudson.

**Software:** Johannes Asplund-Samuelsson.

**Supervision:** Elton P. Hudson.

**Validation:** Johannes Asplund-Samuelsson.

**Visualization:** Johannes Asplund-Samuelsson.

**Writing – original draft:** Johannes Asplund-Samuelsson, Elton P. Hudson.

**Writing – review & editing:** Johannes Asplund-Samuelsson, Elton P. Hudson.

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
