## [Decision Letter · Decision Letter 0]

3 Dec 2020

Dear Dr. Hudson,

Thank you very much for submitting your manuscript "Wide range of metabolic adaptations to the acquisition of the Calvin cycle revealed by comparison of microbial genomes" for consideration at PLOS Computational Biology.

As with all papers reviewed by the journal, your manuscript was reviewed by members of the editorial board and by several independent reviewers. In light of the reviews (below this email), we would like to invite the resubmission of a significantly-revised version that takes into account the reviewers' comments.

We cannot make any decision about publication until we have seen the revised manuscript and your response to the reviewers' comments. Your revised manuscript is also likely to be sent to reviewers for further evaluation.

Please note the Reviewer #1 declined to provide a review. While they thought that the work was very interesting, they did not feel confident providing a review due to lack of technical expertise. In the interest of time, we decided to proceed with two reviews.

Sincerely,

Christopher V. Rao

Associate Editor

PLOS Computational Biology

Jian Ma

Deputy Editor

PLOS Computational Biology

Reviewer's Responses to Questions

**Comments to the Authors:**

**Reviewer #1: **none

**Reviewer #2:** Authors are presenting an extensive bioinformatical analysis of Calvin-Benson cycle and the metabolic adaptation of closely related pathways. The presented study provides valuable information both for other theoretical approaches as well for molecular engineering and biotechnological applications. However, there are several issues which should be addressed:

Major comments:

1) Cyanobacteria diverge strongly from other bacteria due to an ancient evolutionary emergence and were therefore excluded from further analysis.

Differences in adaptation towards Calvin cycle among cyanobacteria themselves, as well as to other photosynthetic bacteria, might be the most interesting results also from the biotechnological point of view. The question is if their analysis would have to be done separately and what would be the control for each species (closest bacterium/cyanobacterium, the simplest/oldest cyanobacterium, …)

2) „Photorespiration salvage did not appear to be adapted specifically for the Calvin cycle.“

Excluded cyanobacteria might have shown a very different picture here. It is known that isozymes have a significant impact on metabolic adaptation, namely the second reaction of photorespiration is producing a toxic compound which is dealt with by 2-4 isozymes of phosphoglycolate phosphatase in cyanobacteria. And that is not the only isozyme within photorespiratory pathway. Authors mentioned isozymes for Calvin-Benson cycle and glycolysis but not for photorespiration; I guess that there are none in other bacteria? This whole study is about contrast and comparison, I cannot imagine a bigger contrast or maybe a surprise if cyanobacteria would have been included in this analysis.

3) Authors picked the presence of large subunit of RuBisCO and Prk as an evidence of Calvin cycle. However, is it just their assumption or undisputed evidence for the existence of fully functional Calvin cycle?

4) Presented algorithm for Calvin cycle identification has an accuracy around 75% and is compared basically to flipping a coin (50/50). It would be more valuable to compare its accuracy to some other algorithm(s) from the same field.

5) Analysis presented on Fig. 5 is missing one of key exit point out of Calvin cycle, the phosphoketolase pathway. It would be interesting to see the importance of this pathway in comparison to lower EMP glycolysis as it prevents the decarboxylation.

6) Rather technical language, paragraphs not linked together, and usage of uncommon synonyms made the reading difficult to follow what authors wanted to say.

Minor comments:

1) „Photorespiration salvage“ – this phrase is not commonly used, probably due to the fact that photorespiratory pathway has other roles than „just“ salvaging phosphoglycolate. If authors want to use it, I would recommend to use quotation marks for the word salvage.

2) Few little grammar errors here and there but nothing uncommon for the manuscript of this length.

**Reviewer #3: **Although improving the efficiency of the Calvin cycle in plants has been a long-term goal for genetic engineers, only modest progress has been made so far. In this work, Asplung-Samuelsson and Hudson try to take a step back and explore the phylogenetic tree for genetic adaptations that have been relevant for this pathway using strong statistical methods. The results of this analysis could prove very valuable for metabolic engineers working on autotrophic growth and therefore might help the global effort to mitigate climate change.

The workflow is well described in the text, and the analysis code is freely available on GitHub, which is commendable.

Major comment:

It’s not very clear why three different methods were used for ranking the features. Presumably, using the consensus ranking of all 3 prevents certain outliers that could come from one of the three methods and increases the confidence of the ranking. Whatever the reason, showing results from different classifiers throughout the text is more confusing than helpful. It would be easier to follow of a single ranking system was used (and justified) throughout the results section. For example, in figure 5, colors indicate RF importance while thickness indicates enrichment score. What about the ACE scores? Why not show the consensus ranking instead (e.g. color coded only)?

In the text itself, ranking from different systems is often mentioned side by side, which is not very helpful. The fact that ACE provides many correlations depending on the sub-tree only adds to the confusion. Also, having so many scores gives the impression that some results are cherry-picked.

Minor comments:

• The definition of “consensus rank” is not given before it is first used in line 171.

• Line 167-169: The Spearman correlation between ACE and random forest is higher than the other two pairs, but the p-value is not lower. I assume it is because there are only 1200 features for the RFs. Perhaps indicating “n” as well as p-value and “r” would help readers wondering about the apparent inconsistency.

• Figure 4: it’s not clear to me why the fact that most archea are CBB- explains the negative correlation with CbbQ.

• Line 250-251: I suggest changed “compared to glycolysis” with something more specific such as “compared to heterotrophic growth on glycolytic carbon sources such as glucose” (FBP and TKT are not even part of glycolysis).

• Figure 5: since the colorbar is scaled to be 0-1 after the log10 transform, the exponent of the log-transform is irrelevant (it is enough to say that it is logarithmic scale).

• Line 324-330: the Entner-Doudoroff pathway does not convert glucose-6-phosphate to ribulose-5-phosphate (Gnd is not part of that pathway). And although Zwf can be considered part of the pathway, it is not exclusive to the ED. I suggest writing about “the ED and OPP pathways” together since they are difficult to distinguish in the context of this analysis.

**Have all data underlying the figures and results presented in the manuscript been provided?**

Reviewer #1: **No: **not read it all supplement

Reviewer #2: Yes

Reviewer #3: Yes

PLOS authors have the option to publish the peer review history of their article (what does this mean?). If published, this will include your full peer review and any attached files.

Reviewer #1: No

Reviewer #2: No

Reviewer #3: **Yes: **Elad Noor
---

## [Editor Report · Decision Letter 1]

25 Jan 2021

Dear Dr. Hudson,

We are pleased to inform you that your manuscript 'Wide range of metabolic adaptations to the acquisition of the Calvin cycle revealed by comparison of microbial genomes' has been provisionally accepted for publication in PLOS Computational Biology.

Best regards,

Christopher V. Rao

Associate Editor

PLOS Computational Biology

Jian Ma

Deputy Editor

PLOS Computational Biology

---

## [Editor Report · Acceptance letter]

4 Feb 2021

PCOMPBIOL-D-20-01954R1 

Wide range of metabolic adaptations to the acquisition of the Calvin cycle revealed by comparison of microbial genomes

Dear Dr Hudson,

I am pleased to inform you that your manuscript has been formally accepted for publication in PLOS Computational Biology. Your manuscript is now with our production department and you will be notified of the publication date in due course.

With kind regards,

Alice Ellingham
